# The Risk of Pediatric Overweight and Children’s Objectively Measured Sedentary Behaviors and Physical Activity by Area of Residence

**DOI:** 10.3390/healthcare13050462

**Published:** 2025-02-21

**Authors:** Aristides M. Machado-Rodrigues, Daniela Rodrigues, Helena Nogueira, Augusta Gama, Helder Miguel Fernandes, Antonio Stabelini Neto, Cristina Padez

**Affiliations:** 1University of Coimbra, Faculty of Sport Sciences and Physical Education, 3040-248 Coimbra, Portugal; 2University of Coimbra, Interdisciplinary Center for the Study of Human Performance (CIPER), 3040-248 Coimbra, Portugal; 3Research Centre for Anthropology and Health, University of Coimbra, 3000-456 Coimbra, Portugal; drdc@uc.pt (D.R.); uc6354@fl.uc.pt (H.N.); maantunes@ciencias.ulisboa.pt (A.G.); cpadez@antrop.uc.pt (C.P.); 4Department of Life Sciences, University of Coimbra, 3000-456 Coimbra, Portugal; 5Department of Geography and Tourism, University of Portugal, 3000-456 Coimbra, Portugal; 6Faculty of Sciences, University of Lisbon, 1749-016 Lisbon, Portugal; 7Polytechnic Institute of Guarda, 6300-559 Guarda, Portugal; hmfernandes@ipg.pt; 8Sport Physical Activity and Health Research & Innovation Center (SPRINT), 6300-559 Guarda, Portugal; 9Health Sciences Center, State University of Northern Parana, Jacarezinho 86400-000, Brazil; asneto@uenp.edu.br

**Keywords:** area of residence, sedentary behavior, obesity, children, physical activity, public health

## Abstract

**Background/Objectives**: Research considering objectively measured moderate-to-vigorous physical activity (MVPA) and sedentary behaviors (SB) and health outcomes among urban children has produced equivocal findings. Therefore, this study was designed (i) to compare MVPA and SB of urban and non-urban children and (ii) to analyze associations between the risk of overweight and MVPA of children by their degree of urbanization. **Methods**: The sample comprised 389 children (195 girls) aged 6 to 10 years. Measurements for height and weight were taken, followed by the calculation of body mass index (BMI). A motion sensor was employed to gather data on MVPA and SB for 7 days. Urban regions were characterized as areas with a population density exceeding 500 inhabitants per square kilometer or having a total population greater than 50,000. ANOVAs, partial correlations, and logistic regression analysis were used, controlling for potential confounders. **Results**: After controlling for wearing time and sex, urban children were significantly less active (lower MVPA) than non-urban peers [i.e., urban: 48 min/day vs. 51 min/day (non-urban)]. On the other hand, urban children spent significantly more time on SB than their non-urban counterparts on the weekend and during all assessed days. MVPA was significantly associated with the risk of being overweight at the weekend for both urban and non-urban children. Inspection of the final regression model indicated that urban children who engaged in sedentary behavior for extended periods were at a higher risk of being categorized as overweight. **Conclusions**: Findings revealed the association between MVPA and the risk of overweight on weekdays was just significant for urban children. The location where a child lives significantly influences their weight status, and therefore, community-based programs [at schools (PE and recess) and after-school (organized sports)] should be developed that encourage active lifestyles tailored to urban environments.

## 1. Introduction

The global prevalence of obesity among children has increased globally, with significant implications for their health. Recent research reveals that around 28% of children from elementary schools are categorized as overweight or obese, which significantly enhances their susceptibility to several comorbidities, including type 2 diabetes, cardiovascular diseases, and even mental health disorders [1]. Even though different methods and scientific approaches are used to assess obesity in pediatric people, all the available data have similarly shown a substantial and rapid increase in the numbers of children affected. An estimated one out of every three children in the European Union states is believed to be overweight or obese, with the highest prevalence observed in southern European and Mediterranean nations [2,3]. In Portugal, according to the World Health Organization (WHO) definition, the results for the last decade (i.e., childhood overweight and obesity in Europe: changes from 2007 to 2017) show a slight reduction in the prevalence of excess weight and obesity in children between the ages of 6 to 8 years has decreased from 37.9% to 29.6% and from 15.3% to 12.0%, respectively [4]. However, this reduction is not linear to all the population, particularly in children from lower socioeconomic status (SES) [5]. Indeed, the WHO highlights that high prevalences are particularly alarming in low- and middle-income countries, where rapid urbanization and shifts in physical activity (PA) and dietary habits foster unhealthy lifestyle choices. Therefore, it is imperative to tackle these escalating obesity rates to prevent long-term health issues and promote a healthier future generation.

The global population residing in urban areas has substantially increased during the 20th century and into the 21st century, primarily in pursuit of improved social and economic living conditions. However, among children and youth, the growth and developmental advantages associated with urban living have significantly diminished in the twenty-first century in many regions worldwide [6]. Among the several biological, behavioral, and environmental factors that influence pediatric health, the residential area where children are living significantly affects their lifestyle choices [7]. This effect manifests through daily habitual PA, time spent in sedentary behaviors, as well as dietary patterns, which have a profound impact on the health of pediatric people. Research findings about the PA levels of children in urban versus non-urban settings have yielded contrasting results. On one hand, some studies indicate that detrimental trends in children’s PA and screen time occur at a higher rate in non-urban areas compared with urban environments [8,9], while other research suggests that rural children participate more frequently in organized sports teams [10]. Additionally, further research has revealed that children residing in urban areas exhibit the lowest levels of PA, whereas those living in smaller cities demonstrate slightly higher activity levels than their urban peers [11]. In fact, the literature has pointed out substantial differences in PA levels between youth from non-urban and urban settings, and complementary studies also indicate that these differences are not just associated with geographic location but also intersect with socioeconomic status [9,12,13,14]. Urban children from lower-income families may have reduced access to safe recreational spaces compared to their higher socioeconomic peers [12]. Therefore, understanding these disparities is crucial for developing targeted interventions aimed at promoting healthy and active lifestyles among children.

Therefore, in the context of the preceding trends, the purpose of the present study was twofold: (i) to compare MVPA levels and sedentary time of urban and non-urban children and assess compliance of active children with the PA recommendation; (ii) to analyze associations between the risk of overweight and MVPA of children by their degree of urbanization. It was hypothesized that urban children would spend more time on sedentary behaviors, and would be less active than their non-urban counterparts, particularly on its MVPA intensity level; in addition, it was hypothesized that the duration of SB would have a positive correlation with the likelihood of being overweight among children aged 6 to 10 years, especially in urban settings.

## 2. Materials and Methods

### 2.1. Participants and Study Design

Children were enlisted as part of the ObesInCrisis project titled “Inequalities in Childhood Obesity: The Impact of the Socioeconomic Crisis in Portugal from 2009 to 2015”. The sampling design for the cross-sectional study conducted in the first half of the year 2017 was the same as in the previous project carried out by the team in 2009–2010, to assess childhood obesity prevalence and the obesogenic environment. [7]. A total of 8472 school-aged children (mean age: 7.2 years; standard deviation: 1.9 years; 49.2% female) were recruited from 118 schools located in the cities of Porto, Coimbra, and Lisbon. Participation rates were 60% in Porto, 58% in Coimbra, and 67% in Lisbon. Details are available elsewhere [15].

The current study represents a distinct segment of the ObesInCrisis project, which involved 395 children (198 girls) drawn from a school population of 5211 students, encompassing all students in grades 1 through 4 residing in the Coimbra district. Children were selected from several aforementioned public schools and were aged between 6 and 10 years, as these specific age groups are critical transitional points for the adoption of healthy lifestyles among Portuguese children. The inclusion criteria of the study were as follows: (i) children of both sexes aged between 6 and 10 years, (ii) school attendance in one of the selected primary schools, and (iii) participation in all anthropometric measures; (iv) wearing the tri-axial accelerometer at least 10 h each day for at least 5 of the 7 assessed days (of which at least 1 day is a weekend day); (v) providing the post code of their area of residence by the parents and/or legal guardian; and (vi) signed informed consent by the parents and/or the legal guardian. Among those 389 children who participated in the current study, 16% of urban children and 21% of non-urban children were overweight or obese. The educational levels of the parents varied, with urban fathers having 68% with the highest education level and 6% with the lowest, and urban mothers having 83% with the highest education level and 3% with the lowest. In contrast, non-urban fathers had 27% with the highest education level and 21% with the lowest, and non-urban mothers had 63% with the highest education level and 9% with the lowest.

Ethical approval for the project was given by the Direção-Geral de Inovação e Desenvolvimento Curricular (Study Registration NO. 0565500003/DGIDC; 28 October 2016), which requires anonymity and non-transmissibility of data. All procedures adhered to the ethical standards set by the Portuguese Data Protection Authority (CNPD, authorization number 745/2017), as well as the 1964 Helsinki Declaration and its later amendments or comparable ethical standards. Moreover, prior to data collection, informed written assent was obtained from parents or guardians.

### 2.2. Variables

#### 2.2.1. Anthropometry

Height and weight measurements were objectively assessed by two trained technicians at the school in the morning, utilizing a portable Seca 217 stadiometer and portable Seca 770 scales, with precision to the nearest 0.1 cm and 0.1 kg, respectively. Participants were attired in t-shirts and shorts and were barefoot during the measurements. Body mass index (BMI, kg/m^2^) was calculated and classified according to age and sex-adjusted cut-off points [16]. The sample was categorized into two weight status groups: normal weight and overweight (which includes both overweight and obesity).

#### 2.2.2. Physical Activity (PA) and Sedentary Behavior (SB) Objectively Assessed

PA and SB were objectively measured for 7 consecutive days using a wGT3X-BT Actigraph accelerometer (ActiGraph LLC, Pensacola, FL, USA). The tri-axial accelerometer was securely positioned on the hip with an elastic belt situated above the right anterior superior iliac spine. The acceleration data collected was filtered and digitized, with the resultant magnitude aggregated over a user-defined time frame (epoch interval), which was established at 5 s. This duration aligns with methodologies employed in previous studies involving children, where it has been demonstrated to yield more precise evaluations of their spontaneous and intermittent activities [17].

The accelerometer data were downloaded electronically using ActiLife 6 software, and the criteria for data processing and inclusion were consistent with those employed in previous European studies, including the European Youth Heart Study [18], the Avon Longitudinal Study of Parents and Children [19], and the Midlands Adolescent Lifestyle Study [13]. Specifically, non-wearing time was characterized by periods of at least 20 consecutive minutes of zero counts. After excluding sequences of 20 or more consecutive zero counts while permitting interruptions of up to 2 min [18,20], any day on which participants failed to record a minimum of 10 h of accelerometer data were omitted from further analyses.

From the initial sample of 415 children, 20 children (5% of participants) failed to reach 10 h of daily wearing time on the measured days, and 6 participants did not fulfill their address/post code to be classified as urban or non-urban; therefore, those children do not meet the criteria for inclusion, and they were not included for subsequent analyses. After applying inclusion criteria, 389 children of both sexes remained in the final analysis of the present study.

For children aged 6 to 10 years, accelerometer data were analyzed using intensity-based cut-off points that classify activity counts into sedentary, light, moderate, or vigorous physical activity. Daily totals of sedentary behavior and moderate-to-vigorous physical activity were reported in minutes per day, calculated with specific pediatric cut-off points [21]. Similarly to previous epidemiological studies [22,23], participants were classified as active if they accumulated at least 60 min of MVPA daily (≥60 min/day of MVPA) and non-active if children did not reach these recommended values (<60 min/day of MVPA).

#### 2.2.3. Area of Residence

Participants were categorized based on their residential area as either urban or non-urban, following the criteria established by the Portuguese Statistical System [24]. Non-urban areas were defined as cities with a population density of fewer than 500 inhabitants per square kilometer or fewer than 50,000 total inhabitants. After applying the inclusion criteria of accelerometry, the other 6 participants were excluded since the parents did not provide the post code of their area of residence.

#### 2.2.4. Parental Education

The educational background of fathers and mothers was assessed by a self-report instrument [5,7], and it was used as a proxy for socio-economic status. It was based on the Portuguese Educational System, respectively: 1 = Low Education (9 years or less–sub-secondary), 2 = Middle Education (10–12 years–secondary), and 3 = High Education (college or university degree). The same procedures have been applied in Portugal [25].

### 2.3. Statistical Procedures

Descriptive statistics were calculated, including means and standard deviations. Normality tests (Kolmogorov–Smirnov) were performed on indicators of habitual PA (counts per minute), MVPA, and SB, revealing that PA and SB measures were normally distributed. Given that participants who are awake longer may have increased opportunities for SB, wearing time was included as a covariate in the analyses of SB and various PA intensity components. Consequently, analyses of covariance (ANCOVA) with wearing time as a covariate were conducted for sex-specific comparisons. Additionally, logistic regression analysis was employed to estimate associations between MVPA and the risk of overweight/obesity while controlling for potential confounding variables such as sex, chronological age, SB, and parental education. In the minimally adjusted model (Model 1), MVPA (minutes/day) was the only predictor of the child’s overweight/obesity risk. Subsequently, sex and chronological age were included as potential confounders in Model 2. Model 3 added sedentary behavior (SB) as another confounding factor, while parental education was incorporated in the final model (Model 4). Significance was set at 5%, and analyses were conducted using SPSS 27.0 (SPSS Inc., Chicago, IL, USA).

## 3. Results

The sample characteristics stratified by area of residence are summarized in Table 1. In urban areas, approximately 79% of girls were classified as normal weight, while 16% were overweight and 5% obese. For urban boys, the figures were 88% normal weight, 9% overweight, and 3% obese. Among non-urban children, about 75% of girls were categorized as normal weight status, 19% as overweight, and 6% as obese; corresponding values for non-urban boys were 82% and 18% for normal weight status and overweight males, respectively.

Urban and non-urban children did not significantly differ in height, weight, and BMI, whereas MVPA on weekdays was significantly lower in urban than in non-urban children, after controlling for sex and wearing time by accelerometer [i.e., urban: 48 min/day vs. 51 min/day (non-urban)]. On the other hand, urban children spent significantly more time on sedentary behavior than their non-urban peers on the weekend and during all assessed days. Furthermore, non-urban children spent significantly more time than their urban counterparts on light activities across all assessed days.

The results indicated a troubling pattern, suggesting that the majority of both male and female participants failed to adhere to the existing physical activity standards for the overall duration of evaluation (i.e., urban active children: 39% males and 22% females; non-urban active children: 59% males; 28% females). In addition, among urban children, the rates of active children showed significant variation between weekdays and weekends for both boys (decreasing from 45% to 31%) and girls (increasing from 23% on weekdays to 28% on weekends). Similarly, among children living in non-urban areas, the activity rates fluctuated notably from weekdays to weekends for boys (dropping from 59% to 36%) and girls (rising from 19% on weekdays to 41% on weekends).

Table 2 presents the partial correlations, controlling for sex and wearing time accelerometry. Results revealed that, among both urban and non-urban children, time spent on MVPA at the weekend was inversely correlated with BMI (i.e., urban children: *r* = −0.12, *p* ≤ 0.05; non-urban children: *r* = −0.33, *p* ≤ 0.01). Furthermore, habitual PA provided by counts/min was also inversely related to the BMI of both urban (*r* = −0.12, *p* ≤ 0.05) and non-urban (*r* = −0.30, *p* ≤ 0.01) children. As expected, the magnitude of those relationships was weak to moderate.

Associations between MVPA and overweight/obesity risk by geographic area of residence, controlling for the aforementioned confounding variables, are presented in Table 3 (on weekdays) and Table 4 (at the weekend). The crude logistic regression model revealed no significant association between MVPA and the risk of being overweight in children of both residential areas during weekdays. However, after controlling for potential confounders, MVPA was associated with the risk of being overweight at the weekend for both urban and non-urban children. Of interest, on weekdays, the association between MVPA and the risk of overweight was just significant for urban children.

Inspection of the final regression model revealed that urban children who spent more time in sedentary behavior were more likely to be classified as overweight on weekdays (OR: (OR = 0.98; 95% CI, 0.984 to 0.975, *p* < 0.01) at the weekend (OR = 0.99; 95% CI, 0.994 to 0.990, *p* < 0.01).

## 4. Discussion

The advantages of living in cities for the healthy growth and development of children and adolescents are diminishing in the 21st century [6]. Considering the current context of urbanization and the substantial variation in the way rurality was defined across studies, as well as the inconsistent results on the differences in PA, sedentary behaviors, and weight status between rural and urban children, further research is still needed to provide a better picture of the lifestyle inequalities among communities from different places of residence. Thus, the present study revealed urban children were significantly less active (i.e., less overall PA and less time devoted to MVPA) on weekdays than their non-urban peers (i.e., urban MVPA: 53.5 min/day vs. non-urban MVPA: 59.3 min/day, *p* < 0.01, η^2^ = 0.011), after controlling for sex and wearing time by accelerometer, corroborating previous studies from the US [9], Australia [26], and Brazil [27]. Indeed, the literature has consistently pointed out substantial differences in PA levels between children from non-urban and urban settings, which are particularly due to environmental, social, and economic influences [24]. In urban areas, adolescents often have access to structured sports programs and a large range of recreational PA facilities, which have promoted their physical engagement. However, the higher population density of urban environments used to lead to important barriers such as traffic congestion and limited green spaces, among others, which have limited opportunities for outdoor play and active commuting of youth (e.g., walking or biking) [28]. In contrast, children from non-urban or rural communities used to experience greater opportunities for unstructured PA due to more open spaces and natural environments conducive to outdoor play, as well as less parental supervision during outdoor activities compared to their urban counterparts [12].

Of note, at higher intensity levels of PA, non-urban children in this study were also revealed to be more active than urban children, who spent significantly less time devoted to MVPA (i.e., less than 6 min of MVPA per day, *p* < 0.01, η^2^ = 0.011); this is undoubtedly interesting, and it is also somewhat contrasting with the features where children are living, as non-urban children may face challenges such as fewer organized sports programs and limited access to sports facilities. This lack of structured programming in non-urban areas might result in lower MVPA levels among children, but it was not observed in the present study. Of interest, the findings of the present study revealed a concerning trend, which pointed out that most of both boys and girls did not meet the current international MVPA guidelines for the total days of assessment (i.e., urban active children: 39% males and 22% females; non-urban active children: 59% males and 28% females). Encouraging greater involvement in sports, which strongly impacts daily energy expenditure and particularly its MVPA intensity portion among youth [13], is also a crucial goal outlined in the Healthy People 2030 Physical Activity objectives for all geographic and diverse density communities [29]. However, attention needs to be paid in some communities because the existing pay-to-play structure of the youth sports system has created significant barriers to participation, disproportionately affecting children from minority backgrounds. This has unveiled further social and economic disparities among children, since youth from families with higher income levels revealed significantly higher levels of organized sports participation than their low-income peers [30], indicating that the aforementioned differences in the PA levels are not solely attributable to geographic location but also intersect with socioeconomic status.

On the other hand, the present study shows complementary trends in the urban vs. non-urban lifestyles contrast, revealing that urban children spent significantly more time on sedentary activities than their non-urban peers at the weekend (i.e., urban SB: 462.5 min/day vs. non-urban SB: 439.8 min/day, *p* < 0.04, η^2^ = 0.011), and during all assessed days (i.e., urban SB: 508.4 min/day vs. non-urban SB: 503.0 min/day, *p* < 0.01, η^2^ = 0.016). Indeed, sedentary behavior among children has been a growing concern, particularly when comparing rural and urban settings. Studies reveal that urban children generally exhibit higher levels of sedentary behavior than those in non-urban areas of residence [31,32]. This discrepancy can be attributed to several factors, including access to recreational facilities, transportation options, and lifestyle choices. Urban locales typically offer more chances for sedentary pursuits, such as increased screen time, due to the widespread availability of technology and limited outdoor spaces for PA [33]. Moreover, socioeconomic factors play a significant role in these patterns since urban children may also have greater access to digital devices and entertainment options that promote sedentary lifestyles. Of note, the present study also revealed that urban children who spent more time in sedentary activities were more likely to be classified as overweight on weekdays (OR: (OR = 0.98; 95% CI, 0.984 to 0.975, *p* < 0.01) at the weekend (OR = 0.99; 95% CI, 0.994 to 0.990, *p* < 0.01), calling for further detailed studies of different types of screen activities and their specific and independent impact on the weight status of younger children. Indeed, recognizing these differences is vital for crafting specific interventions aimed at decreasing sedentary behavior among children, especially those from urban settings.

Pediatric obesity has a complex etiology, and therefore, it is imperative to identify behavioral, social, and environmental causes for subsequently designing effective strategies to overcome that public health concern. The literature on the contrasts of geographic areas of residence has highlighted the variations in PA as a significant element that could account for the increased rates of obesity found among youth residing in urban environments compared to those in rural or non-urban areas [9,27,34]. The results of the current research indicated that the correlation between MVPA and the likelihood of being overweight or obese during weekdays was only significant among children living in urban areas. Similar results were obtained from a recent meta-analysis of 10 studies examining urban and rural differences in childhood obesity in the United States, revealing that among a combined sample of over 74,000 children aged 2 to 19 years, those residing in rural regions exhibited a 26% higher likelihood of being obese compared to their urban counterparts [35]. Further studies found a similar difference, with youth from non-urban areas having 28% greater odds of obesity compared with those from urban and/or metropolitan areas of residence [36]. Indeed, non-urban areas frequently experience higher rates of food insecurity, and they are more likely to be classified as food deserts, which limits the availability of nutritious foods necessary for maintaining a healthy weight [37]. The prevalence of poverty is also notably higher in rural regions, contributing to stressors that can lead to reduced opportunities for active lifestyles and unhealthy eating habits [38]. For example, further studies suggested that rates of fruit and vegetable consumption were lower among rural youth than among suburban and urban youth [39]. These interconnected issues create an environment where rural children face greater risks for overweight and obesity compared to those living in urban settings, claiming for additional effective strategies.

The current research indicated that the geographic settings in which individuals reside (whether in rural or urban areas of residence) may significantly influence the promotion of active lifestyles among children of younger ages. A notable strength of this investigation, in contrast to earlier studies, was its analysis of the variations in the associations between different intensity levels of PA objectively assessed and the risk of being overweight during both weeks and weekends of children living in urban versus rural communities. Conversely, the present study is not without limitations, which have to be acknowledged. First, the findings are exclusively for children aged 6 to 10 years who live in the Portuguese Midlands. The built environments of the communities were not considered, and the associations between BMI and PA may also vary with the culture or socioeconomic status of families. Second, results should be analyzed with caution because cut-offs for intensity levels of PA are a work-in-progress issue, with accelerometer calibration data being added to the scientific evidence once values become available [40,41]. In addition, the relationship between PA/SB and area of residence may be influenced by other factors not considered in the study, e.g., distance from home to school, availability of PA facilities, attitudes towards physical activity, and typology of spontaneous play, as well as social inequalities not reflected in parental education and perhaps subtle cultural differences among youth. Furthermore, although accelerometers provide an objective and reasonably accurate measure of PA, the instrument does not capture specific activities, e.g., manual and water activities, and it fails to encompass all dimensions of PA, such as activities involving scenarios where wearing the accelerometer poses a physical danger. Thus, given that the accelerometer is not considered the “gold standard” for evaluating habitual PA or MVPA, and that self-reported instruments heavily rely on participants’ memory and past experiences with PA, it is advisable for future research involving children to adopt a multi-method approach that combines both quantitative and qualitative data on PA and SB. On the other hand, it does not account for certain sedentary behaviors (SB), including variations in sleep duration. Furthermore, body mass index (BMI) is the most utilized measure of overweight and obesity in epidemiological research; however, it may not adequately reflect variations in body composition that are pertinent to physical activity and cardiorespiratory fitness (CRF). However, as previously mentioned, positive experiences of PA in different settings and dimensions, even for those with higher body fatness, are important strategies for encouraging active lifestyles and mitigating the adverse effects of obesity on community health.

To enhance the practical implications of the present study for child health promotion, particularly in urban settings where children exhibit lower levels of MVPA and higher SB, it is crucial to adopt a multifaceted approach. First, community-based programs should be developed that encourage active play and physical activities tailored to urban environments, such as diverse organized sports programs or after-school programs that promote structured active behavior. Second, schools should integrate more physical education into their curricula and create safe spaces for children to engage in physical activities during recess. Third, public health campaigns must focus on educating families about the importance of reducing screen time and promoting active lifestyles, particularly during weekends when SB is notably higher. Additionally, collaboration with local governments to improve urban infrastructure—such as creating more parks and safe walking paths—can facilitate increased physical activity among children. Finally, ongoing research should be conducted to monitor the effectiveness of these interventions and adapt them based on the unique needs, particularly for those with higher-density people.

## 5. Conclusions

Findings of the present study revealed that after controlling for wearing time and sex, urban children were significantly less active (lower MVPA) than non-urban peers (i.e., urban MVPA: 53.5 min/day vs. non-urban MVPA: 59.3 min/day, *p* < 0.01, η^2^ = 0.011). Furthermore, urban children engaged in considerably greater amounts of SB compared to their rural peers during weekends and throughout all evaluated days. The inverse association between MVPA and the risk of being overweight on weekdays was just significant for urban children. Therefore, the location where a child lives significantly influences their weight status, emphasizing the importance of comprehending the intricacies of their daily life in urban and rural communities to better delimit educational and possibly clinical interventions for public health promotion; thus, community-based programs [at schools (physical education classes and recess) and after-school (organized sports)] should be developed that encourage active lifestyles tailored to urban environments.

## Figures and Tables

**Table 1 healthcare-13-00462-t001:** Descriptive statistics and results of ANCOVAs (controlling for sex and total measured time by accelerometry) testing the effect of degree of urbanization on body size, sedentary behavior, and different intensity portions of Physical Activity (PA) in children (*n* = 389).

Variables	Urban (*n* = 318)	Non-Urban (*n* = 71)	*p*	η^2^
Chronological age, years	8.4 ± 1.2	8.4 ± 1.1	0.85	0.000
Height, cm	128.9 ± 9.1	128.8 ± 8.3	0.91	0.000
Weight, kg	28.4 ± 7.4	28.6 ± 7.0	0.79	0.000
BMI, kg/m^2^	16.88 ± 2.73	17.00 ± 2.39	0.65	0.001
PA (weekdays), counts/min	462.0 ± 136.3	495.8 ± 134.5	0.12	0.006
PA (weekend), counts/min	440.1 ± 201.6	471.8 ± 279.7	0.37	0.002
PA (total of 7 days), counts/min	455.7 ± 133.0	489.0 ± 145.1	0.12	0.006
SED (weekdays), min/day	526.8 ± 100.9	528.3 ± 87.1	0.52	0.001
SED (weekend), min/day	462.5 ± 160.1	439.8 ± 144.2	0.04	0.011
SED (total of 7 days), min/day	508.4 ± 98.5	503.0 ± 88.4	0.01	0.016
Light PA (weekdays), min/day	178.3 ± 37.7	192.2 ± 39.3	0.01	0.019
Light PA (weekend), min/day	171.4 ± 59.5	176.4 ± 69.7	0.78	0.000
Light PA (total of 7 days), min/day	176.3 ± 36.5	187.7 ± 40.6	0.02	0.014
MVPA (weekdays), min/day	53.5 ± 18.5	59.3 ± 19.2	0.04	0.011
MVPA (weekend), min/day	48.4 ± 25.3	51.0 ± 30.0	0.66	0.001
MVPA (total of 7 days), min/day	52.0 ± 17.9	57.0 ± 18.3	0.08	0.008

Note: BMI (body mass index); PA (physical activity); SED (minutes spent sedentary); MVPA (moderate-to-vigorous physical activity).

**Table 2 healthcare-13-00462-t002:** Partial correlations (controlling for sex and measured time by accelerometry) between BMI and time spent sedentary and physical activity in its different intensities by geographic contexts.

Variables	BMI	
	Urban	Non-Urban
	(*n* = 318)	(*n* = 71)
PA (weekdays), counts/min	−0.07	0.17
PA (weekend), counts/min	−0.12 *	−0.30 **
PA (total of 7 days), counts/min	−0.11 (*p* = 0.06)	−0.06
SED (weekdays), min	−0.02	0.12
SED (weekend), min	0.03	−0.22
SED (total of 7 days), min	0.00	−0.04
Light PA (weekdays), min	0.13 *	0.29 *
Light PA (weekend), min	−0.09	−0.25 *
Light PA (total of 7 days), min	0.06	0.09
MVPA (weekdays), min	−0.08	0.12
MVPA (weekend), min	−0.12 *	−0.33 **
MVPA (total of 7 days), min	−0.11 *	−0.07

Note: * *p* < 0.05; ** *p* < 0.01; BMI (body mass index); PA (physical activity); SED (minutes spent sedentary); MVPA (moderate-to-vigorous physical activity).

**Table 3 healthcare-13-00462-t003:** The association between moderate-to-vigorous physical activity (MVPA) with the risk of obesity by geographic context (i.e., urban vs. non-urban residential area) on weekdays.

			Overweight/Obesity Risk				
	*n*	Model ^a^	*B*	S.E.	*e^B^*	95% CI	*p*
Urban context	*n* = 318	1	−0.011	0.010	0.989	0.97 to 1.01	0.28
		2	−0.010	0.011	0.990	0.97 to 1.01	0.38
		3	−0.036	0.015	0.964	0.94 to 0.99	0.02
		4	−0.039	0.015	0.962	0.93 to 0.99	0.01
Non-urban context	*n* = 71	1	0.014	0.016	1.014	0.98 to 1.05	0.39
		2	0.018	0.019	1.018	0.98 to 1.06	0.36
		3	−0.012	0.030	0.988	0.93 to 1.05	0.68
		4	−0.002	0.032	0.998	0.94 to 1.06	0.95

^a^ Model 1 = unadjusted; Model 2 = Model 1 + sex, chronological age, and measured time of accelerometer; Model 3 = Model 2 + time spent sedentary; Model 4 = Model 3 + parental education.

**Table 4 healthcare-13-00462-t004:** The association between moderate-to-vigorous physical activity (MVPA) with the risk of obesity by geographic context (i.e., urban vs. non-urban residential area) at the weekend.

			Overweight/Obesity Risk				
	*n*	Model ^a^	*B*	S.E.	*e^B^*	95% CI	*p*
Urban context	*n* = 318	1	−0.001	0.007	0.997	0.98 to 1.01	0.71
		2	−0.006	0.008	0.994	0.98 to 1.01	0.45
		3	−0.020	0.010	0.980	0.96 to 0.99	0.04
		4	-0.018	0.010	0.982	0.96 to 1.00	0.06
Non-urban context	*n* = 71	1	−0.028	0.014	0.972	0.95 to 0.99	0.04
		2	−0.034	0.017	0.967	0.94 to 1.00	0.05
		3	−0.042	0.021	0.959	0.92 to 1.00	0.05
		4	−0.044	0.021	0.957	0.92 to 0.99	0.03

^a^ Model 1 = unadjusted; Model 2 = Model 1 + sex, chronological age, and measured time of accelerometer; Model 3 = Model 2 + time spent sedentary; Model 4 = Model 3 + parental education.

## Data Availability

The data presented in this study are available on request from the corresponding author.

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
