# Peer review of "The Risk of Pediatric Overweight and Children’s Objectively Measured Sedentary Behaviors and Physical Activity by Area of Residence"

_healthcare, 2025, doi:10.3390/healthcare13050462_

Round 1
Reviewer 1 Report
Comments and Suggestions for Authors
1.The study mentioned socioeconomic factors but failed to deeply explore how various intersecting variables, such as household income, parental employment status, and local recreational facilities, impact PA participation. A more nuanced investigation into these socioeconomic disparities could provide insights into the differing experiences of children across various backgrounds.
2.While the study primarily examined moderate-to-vigorous physical activity (MVPA) and general sedentary behavior, it did not consider other critical forms of activity, such as active commuting or play. Including these dimensions would create a more comprehensive picture of total daily activity levels and their implications for health outcomes.
3.The reliance on accelerometers to measure PA may introduce certain biases, as these devices may not capture all forms of activity (e.g., swimming) or sedentary behaviors (e.g., technology use). This limitation could affect the comprehensiveness of the study, highlighting the need for complementary data collection methods.
4.This article discusses issues and disparities without proposing actionable interventions or programs tailored to address identified concerns. Concrete strategies are necessary to translate the research findings into practical applications aimed at improving children's activity levels and overall health.
5.The binary classification of urban versus non-urban children in this study may oversimplify the complexities within these categories. Variations within urban and rural settings can yield different trends, necessitating a more granular analysis that acknowledges these subtleties.
Author Response
2025 January, 26th
Manuscript ID: healthcare-3407180, entitled " The risk of pediatric overweight and its association with the children’ objectively measured physical activity and sedentary behaviors by area of residence"
The authors thank the reviewers for their thoughtful and construction comments and suggestions. The comments were carefully read, and the revised manuscript was performed accordingly. Responses to specific comments are indicated below in BLUE font.
REVIEWER #1
REVIEWER #1:
1.The study mentioned socioeconomic factors but failed to deeply explore how various intersecting variables, such as household income, parental employment status, and local recreational facilities, impact PA participation. A more nuanced investigation into these socioeconomic disparities could provide insights into the differing experiences of children across various backgrounds.
AUTHORS:
As the reviewer clearly knows science is a rigorous, systematic endeavor that constructs and organizes knowledge in the form of testable explanations and predictions about our surroundings, according with our scientific experience, knowledge and scientific interests. Thus, one of the main drivers of the advancement of science is human diversity and curiosity, uncommitted to concrete results and free from any type of tutelage or rigid standard statement.
Thus, according to the existing literature on this specific field, as well as taking into account the scientific experience of our research team, the authors really would like to clarify that the main purpose of the study was twofold: i) to compare MVPA levels and sedentary time of urban and non-urban children and assess compliance of active children with the PA recommendation; ii) to analyze associations between the risk of overweight and MVPA of children by their degree of urbanization.
Since the afore-mentioned constructs interact together with other biological and behavioral factors, as well as might be mediated by different social, economic and, even, geographic factors, the authors tried to develop a rational and pragmatic analytical approach which would be robust and based on the scientific literature.
Therefore, considering our large implemented projects and our previous scientific strategy, the assessed and available variables related to the socio-economic factors of families were included in the statistical models of the present paper such as parental education which was used as a proxy variable for socio-economic status (SES) – we are completely aware there are several others, probably more precise, but they were not assessed in the present study.
The authors are also completely aware that rigorous and consistent procedures have to be used, as well as there is no perfect research or studies without limitations. Thus, at the methods section, the authors have already mentioned “The educational background of fathers and mothers was used as a proxy for socio-economic status.”.
Since the SES in the present study was not one main construct to be studied, but it was a co-variate instead, the authors kept that clarification throughout the manuscript, as well as discussed it at the limitation section of the study; thus, the manuscript was edited accordingly.
REVIEWER #1:
2.While the study primarily examined moderate-to-vigorous physical activity (MVPA) and general sedentary behavior, it did not consider other critical forms of activity, such as active commuting or play. Including these dimensions would create a more comprehensive picture of total daily activity levels and their implications for health outcomes.
AUTHORS:
The authors agree with the reviewer that other forms or dimensions of physical activity (PA) have also crucial role throughout the childhood. However, the main dimension and portion which has stronger impact on children’s health is the MVPA portion; this is for that that it was studied, particularly using objective measures which confer higher pertinence and rigorous. Instead, active communing has to assessed by self-report and, therefore, it is associated a greater bias, as well as the active play which closely connected to the previous experiences of kids and their families. In addition at this early ages, children are clearly less independent to choose go to school actively, therefore, rates of active commuting at this ages are residual and less relevant.
As the reviewer is completely aware the large multi-dimensional projects are planned by multi-disciplinary research teams in the previous years of the field work and clearly precedent to its dissemination. Therefore, in the present project the authors have clarified the methodological procedures and timely stages, recognizing the science may go further by refining previous procedures.
Thus, the authors have, yet, added some limitations and/or suggestions for future studies and global projects by incorporating further instruments to assess habitual PA, as well as suggest multi-methods approach for future scientific projects of children.
REVIEWER #1:
3.The reliance on accelerometers to measure PA may introduce certain biases, as these devices may not capture all forms of activity (e.g., swimming) or sedentary behaviors (e.g., technology use). This limitation could affect the comprehensiveness of the study, highlighting the need for complementary data collection methods.
AUTHORS:
The authors thank and agree with the reviewer. Therefore, the limitation section of the manuscript was edited accordingly by highlighting some inaccuracy of accelerometer to assess specific tasks and behavioral of the daily lifestyle of children. Thus, the following content was added:
“The relationship between physical activity/SB and area of residence may be influenced by other factors not considered in the study, e.g., distance from home to school, availability of physical activity facilities, attitudes towards physical activity and typology of spontaneous play, as well as social inequalities not reflected in parental education, and perhaps subtle cultural differences among youth. Furthermore, although accelerometers provide an objective and reasonably accurate measure of PA, the instrument does not capture specific activities, e.g., manual and water activities. Thus, given that the accelerometer is not considered the “gold standard” for evaluating habitual PA or MVPA, and that self-reported instruments heavily rely on participants’ memory and past experiences with PA, it is advisable for future research involving children to adopt a multi-method approach that combines both quantitative and qualitative data on PA and SB.”
REVIEWER #1:
4.This article discusses issues and disparities without proposing actionable interventions or programs tailored to address identified concerns. Concrete strategies are necessary to translate the research findings into practical applications aimed at improving children's activity levels and overall health.
AUTHORS:
The authors really thank and agree with the reviewer that practical strategies need to be added for future interventions. Therefore, the manuscript was edited accordingly, as follows:
“To enhance the practical implications of the present study for child health promotion, particularly in urban settings where children exhibit lower levels of MVPA and higher SB, it is crucial to adopt a multifaceted approach. First, community-based programs should be developed that encourage active play and physical activities tailored to urban environments, such as diverse organized sports programs or after-school programs that promote structured active behavior. Second, schools should integrate more physical education into their curricula and create safe spaces for children to engage in physical activities during recess. Third, public health campaigns must focus on educating families about the importance of reducing screen time and promoting active lifestyles, particularly during weekends when SB is notably higher. Additionally, collaboration with local governments to improve urban infrastructure—such as creating more parks and safe walking paths—can facilitate increased physical activity among children. Finally, ongoing research should be conducted to monitor the effectiveness of these interventions and adapt them based on the unique needs, particularly on those with higher density people.”
Thank you very much.
REVIEWER #1:
5.The binary classification of urban versus non-urban children in this study may oversimplify the complexities within these categories. Variations within urban and rural settings can yield different trends, necessitating a more granular analysis that acknowledges these subtleties.
AUTHORS:
The authors thank the reviewer for their thoughtful comments. In fact, in some abroad specific geographic settings the binary classification of urban versus non-urban children may oversimplify the complexities within these categories. However, in the Portuguese context that binary classification is clearly and commonly used; in addition, there were just small societal changes throughout the last decades, and as it is one of the main topics of our research group, we have kept the several conceptual and methodological approach, which are similar to recent national and international publications of our research group as follows:
REFERENCES:
Machado-Rodrigues AM, Rodrigues D, Gama A, Nogueira H, Fernandes HM, Stabelini Neto A, Fernandes R, Padez C. (2025). Can the Urban Lifestyle Impact on Children BMI and Healthy Sleep? Am J Hum Biol. 2025 Jan;37(1): e24210. doi: 10.1002/ajhb.24210. [Impact Factor: 1.60]
NCD Risk Factor Collaboration (NCD-RisC). Diminishing Benefits of Urban Living for Children and Adolescents’ Growth and Development. Nature 2023, 615, 874–883, doi:10.1038/s41586-023-05772-8. [Impact Factor: 69.504]
Machado-Rodrigues AM, Coelho e Silva MJ, Mota J, Padez C, Cumming SP, Malina RM (2014). Urban-rural contrasts in cardio-respiratory fitness, physical activity, and time spent sedentary in Portuguese adolescents. Health Promotion International (Oxford University Press). 2014 Mar;29(1):118-29. doi: 10.1093/heapro/das054. Epub 2012 Oct 19. [Impact Factor: 2.30] [ISI] [ISSN 0957-4824].
Machado-Rodrigues AM, Coelho e Silva MJ, Padez C., Ronque ER, Mota J, Cumming SP, Malina RM (2012). Relationships between Obesity, Cardiorespiratory Fitness, Sedentary Behaviour and Objective Intensity levels of Physical Activity in rural and urban Portuguese adolescents. Journal of Child Health Care, vol. 16, 2: pp. 166-177. doi: 10.1177/1367493511430676 [Impact Factor: 1.896] [ISI, Web of Knowledge] [ISSN: 1367-4935]
Machado-Rodrigues AM, Coelho e Silva MJ, Mota J, Cumming SP, Riddoch CJ, Malina RM (2011). Correlates of aerobic fitness in urban and rural Portuguese adolescents. Annals of Human Biology, Vol. 38(4): 479-484. doi: 10.3109/03014460.2011.554865. [Impact Factor: 1.713] [ISI, Web of Knowledge] [ISSN: 0301-4460]
The authors also thank the reviewer for his careful reading and suggestions which were considered and indubitably contribute to make the manuscript clearer and in a better standard for the potential reader.
Thank you very much.
Reviewer 2 Report
Comments and Suggestions for Authors
Dear Authors,
The research paper you presented addresses an extremely important public health issue, namely pediatric obesity, and provides a valuable analysis of the influence of environmental factors on children’s health. It is commendable that your study emphasizes the need for tailored interventions to promote an active lifestyle according to the geographical location and specific characteristics of the community. Furthermore, it highlights the importance of well-structured educational and public health strategies.
However, I believe there are some shortcomings in the paper that I would like to mention as follows:
1. Title:
The title is overly long and should be reformulated. While it is clear, it could be condensed to emphasize the essence of the study. For example, the title could be: “The Influence of Physical Activity and Sedentary Behavior on Pediatric Obesity: A Comparative Analysis Between Urban and Rural Settings.”
2. Abstract:
The statements regarding the correlations between variables and their significance are not supported by numerical values. Please address this. Additionally, I suggest reformulating the conclusion to include practical implications.
3. Keywords:
The term “active behavior” is too general. It could imply intellectual/cognitive, emotional, motor activities, etc. Consider replacing it with “physical activity,” which is an established term in the field. Additionally, consider adding terms like “sedentary behavior” and “urban vs. non-urban (rural)” as they are defining aspects of this study.
4. Introduction:
o In lines 42–43, consider presenting the figure of 38 million children in percentages to better convey the magnitude of the phenomenon.
o In line 48, the phrase “An estimated one out of every three children in Europe is believed...”—are you referring to Europe as a continent or the European Union states? Please clarify.
o In lines 50 and 55, are you referring to the same WHO document? What year? Please add this reference to the bibliography.
o The influence of parental educational level on pediatric obesity is not sufficiently developed. Please elaborate on this.
5. Participants and Study Design:
o The inclusion and exclusion criteria are not explicitly stated.
o In line 107, “Children with missing information on accelerometry were excluded from the sample”—does this refer to children who could not use accelerometers? Please clarify.
o How were data on parental educational levels collected?
o Nothing is mentioned about the research team.
o When were the data collected? During what period? If this study is derived from one conducted in 2016–2017, i.e., seven years ago, this should be clearly explained, as the information is currently confusing.
6. Area of Residence:
o In lines 164–166, how many participants remained in the final analysis? If the inclusion and exclusion criteria had been clearly defined, this confusion might have been avoided.
7. Results:
o In lines 246–248, if the data are not presented in tables, how can this statement be substantiated?
o There is no connection made between parental education and children’s physical activity. For example, you could interpret the data with a sentence like: “Children whose parents had higher education levels exhibited greater time in moderate-to-vigorous physical activity, p = ..., r = ....”
8. Discussion:
Please include numerical data when stating that certain variables are significantly higher or lower.
9. Conclusions:
The conclusions are general and could benefit from specific direction. For example: “Interventions to promote physical activity should be tailored to the characteristics of each residential environment, taking into account the limited time spent in moderate-to-vigorous physical activity observed in urban children.” Additionally, a statement about the practical implications of this study should be added. For instance: “The results of this study could guide the development of customized school physical education programs.”
Author Response
2025 January, 26th
Manuscript ID: healthcare-3407180, entitled " The risk of pediatric overweight and its association with the children’ objectively measured physical activity and sedentary behaviors by area of residence"
The authors thank the reviewers for their thoughtful and construction comments and suggestions. The comments were carefully read, and the revised manuscript was performed accordingly. Responses to specific comments are indicated below in BLUE font.
REVIEWER #2
REVIEWER #2:
The research paper you presented addresses an extremely important public health issue, namely pediatric obesity, and provides a valuable analysis of the influence of environmental factors on children’s health. It is commendable that your study emphasizes the need for tailored interventions to promote an active lifestyle according to the geographical location and specific characteristics of the community. Furthermore, it highlights the importance of well-structured educational and public health strategies.
AUTHORS:
The authors really would like to thank the reviewer for their thoughtful and construction comments and suggestions. The comments were carefully read, and the revised manuscript was performed accordingly.
REVIEWER #2:
However, I believe there are some shortcomings in the paper that I would like to mention as follows:
- Title:
The title is overly long and should be reformulated. While it is clear, it could be condensed to emphasize the essence of the study. For example, the title could be: “The Influence of Physical Activity and Sedentary Behavior on Pediatric Obesity: A Comparative Analysis Between Urban and Rural Settings.”
AUTHORS:
The authors thank the reviewer for the suggestion, and they have slighted reformulated the title without lose the overall message (i.e. nature of the constructs, its assessment and related analytical approach) equalizing the 19 words title suggestion. Of note, the analyses of the present study are not just comparative but also an associative analysis (of the greater importance).
REVIEWER #2:
- Abstract:
The statements regarding the correlations between variables and their significance are not supported by numerical values. Please address this. Additionally, I suggest reformulating the conclusion to include practical implications.
AUTHORS:
The authors would like to thank the reviewer for the careful reading of the paper; thus, the correlations were double checked, and the results are correct since they are referring to the logistic analysis; thus, the partial correlations as they are a collateral tool of assessment, the authors have just emphasized the main results of the study association constructs provided by the logistic regression analysis (i.e. more robust and powerful statistical tool). In addition, the authors agree with the reviewer and the practical implications of the study were added to this manuscript.
REVIEWER #2:
- Keywords:
The term “active behavior” is too general. It could imply intellectual/cognitive, emotional, motor activities, etc. Consider replacing it with “physical activity,” which is an established term in the field. Additionally, consider adding terms like “sedentary behavior” and “urban vs. non-urban (rural)” as they are defining aspects of this study.
AUTHORS:
Thank you for the note. The authors have edited the paper accordingly by adding the keywords “physical activity” and “sedentary behavior”.
REVIEWER #2:
- Introduction:
In lines 42–43, consider presenting the figure of 38 million children in percentages to better convey the magnitude of the phenomenon.
AUTHORS:
The authors added the corresponding percentage as requested.
REVIEWER #2:
In line 48, the phrase “An estimated one out of every three children in Europe is believed...”—are you referring to Europe as a continent or the European Union states? Please clarify.
AUTHORS:
Good point. That sentence was clarified as requested. Thus, the following content was added: “…one out of every three children in European Union states is believed...”.
REVIEWER #2:
In lines 50 and 55, are you referring to the same WHO document? What year? Please add this reference to the bibliography.
AUTHORS:
The authors have double checked the reference which it is correct (we are referring to the WHO criteria/definition which it is worldwide disseminated). Of note, the cited reference (Buoncristiano et al.; 2021) is a paper from one head-members from the European Office of the WHO and, therefore, it was cited as an original reference [i.e. data are from the WHO European Childhood Obesity Surveillance Initiative (COSI) fourth round (2015–2017)].
REVIEWER #2:
The influence of parental educational level on pediatric obesity is not sufficiently developed. Please elaborate on this.
AUTHORS:
The authors really would like to clarify that the main purpose of the study was twofold: i) to compare MVPA levels and sedentary time of urban and non-urban children and assess compliance of active children with the PA recommendation; ii) to analyze associations between the risk of overweight and MVPA of children by their degree of urbanization.
Since the afore-mentioned constructs interact together with other biological and behavioral factors, as well as might be mediated by different social, economic and, even, geographic factors, the authors tried to develop a rational and pragmatic analytical approach which would be robust and based on the scientific literature.
Of note, the SES in the present study was not one main construct to be studied, but it was a co-variate instead, the authors kept that clarification throughout the manuscript. In addition, the pragmatism and focus on those aims needed to be kept, especially to emphasize the pertinence of the research question, as well as trying to overcome the gap on this specific scientific field of the urban vs. no-urban contrast.
Meanwhile, some discussion was provided by recognizing some limitations of the present study and also by suggesting future strategies both for research and for behavioral intervention on children.
REVIEWER #2:
- Participants and Study Design:
The inclusion and exclusion criteria are not explicitly stated.
AUTHORS:
Thank you for the note. The including and/or excluding criteria were clarified in the methods section as follows:
“The inclusion criteria of the study were as follows: (i) children of both sexes aged between 6 and 10 years, (ii) school attendance in one of the selected primary schools, and (iii) participate in all anthropometric measures; iv) wear the tri-axial accelerometer at least 10 hour each per day for at least 5 of the 7 assessed days (of which at least 1 is a weekend day); v) provide the post code of their area of residence by the parents and/or legal guardian; vi) signed informed consent by the parents and/or the legal guardian. “
REVIEWER #2:
In line 107, “Children with missing information on accelerometry were excluded from the sample”—does this refer to children who could not use accelerometers? Please clarify.
AUTHORS:
Thank you very much for your careful reading. Thus, all inclusion/exclusion criteria and corresponding content was refined throughout the manuscript by providing a clearer information for the potential reader of this paper.
REVIEWER #2:
How were data on parental educational levels collected?
AUTHORS:
That information was added to the sub-heading 2.3.4. (Parental education) with similar studies and references; thus, parental education levels were collected by self-report, similarly to previous epidemiological studies (Machado-Rodrigues et al., 2018; Rodrigues et al., 2020).
Rodrigues, D., A.M. Machado-Rodrigues, and C. Padez, Parental misperception of their child's weight status and how weight underestimation is associated with childhood obesity. Am J Hum Biol, 2020. 32(5): p. e23393.
Machado-Rodrigues, A.M.; Fernandes, R.; Gama, A.; Mourão, I.; Nogueira, H.; Rosado-Marques, V.; Padez, C. The Association of Irregular Sleep Habits with the Risk of Being Overweight/Obese in a Sample of Portuguese Children Aged 6-9 Years. Am J Hum Biol 2018, 30, e23126, doi:10.1002/ajhb.23126.
REVIEWER #2:
Nothing is mentioned about the research team.
AUTHORS:
Specific and the essential details of data collection are presented to the methos section of the paper. For example, it was mentioned that anthropometric variables were collected by experienced / trained researchers from the scientific team. In addition, the author’s contribution is already added at the end of the manuscript (before the references), Probably, it could be added some additional information, but we think it might be clearly collateral like the main research interests of the first and the last author of this manuscript which are as follows:
Aristides Machado-Rodrigues
Citations: 10.094 (scopus)
indexed papers: 106
h-index: 21
https://www.scopus.com/authid/detail.uri?authorId=36519229300
orcid : 0000-0002-7169-8034
Cristina Padez:
indexed papers: 142
h-index: 34
Citations: 17.708 (scopus)
ORCID: 0000-0002-1967-3497
https://www.scopus.com/authid/detail.uri?authorId=6602450500
REVIEWER #2:
When were the data collected? During what period? If this study is derived from one conducted in 2016–2017, i.e., seven years ago, this should be clearly explained, as the information is currently confusing.
AUTHORS:
The data collection was collected in the first half of the year of 2017.The information was clarified throughout the manuscript as follows: “The sampling design for the cross-sectional study conducted in the first half of the year of 2017 was the same as in the previous project carried out by the team in 2009–2010, to assess childhood obesity prevalence and the obesogenic environment.”
Hopefully there is any confusion about this kind of large projects which are designed to analyze large range of constructs/variables and time periods such as decennial trends and its dissemination just can be done later on. Furthermore, as the reviewers are certainly aware, changes in geographic settings and variations in behavioral lifestyles occurs in still larger periods of time, as well as the aforementioned contrasts on urban vs. no-urban features and/or the place where families are living, clearly documented in the cited literature.
REVIEWER #2:
- Area of Residence:
In lines 164–166, how many participants remained in the final analysis? If the inclusion and exclusion criteria had been clearly defined, this confusion might have been avoided.
AUTHORS:
That information is already clarified at the methods section of the manuscript.
Thank you very much.
REVIEWER #2:
- Results:
In lines 246–248, if the data are not presented in tables, how can this statement be substantiated?
AUTHORS:
The authors understand the reviewer’s concern. Once that result is important, but it is not the main result of the study, the authors decided to keep the Tables 3 and Table 4 with that clearer content/data. However, we follow the reviewer suggestion by added that important result in the text of the paper, with more detailed information (i.e. Odds ratio and Confidence Interval, as well as the level of significance on the association between those constructs/variables), as follows:
“Inspection of the final regression model revealed that urban children who spent more time in sedentary behavior were more likely to be classified as overweight on weekdays (OR: (OR = 0.98; 95% CI, 0.984 to 0.975, p<0.01) at the weekend (OR = 0.99; 95% CI, 0.994 to 0.990, p<0.01).”
Thus, there have never been any problems regarding this type of issue/topic, and that aforementioned procedure is massively used and common in this kind of scientific articles, particularly with this type of scientific/methodological approaches.
Finally, as all reviewers are aware, all data are available and mentioned in the paper as follows: “Data Availability Statement: The data that support the findings of this study are available from the corresponding author upon reasonable request.”.
REVIEWER #2:
There is no connection made between parental education and children’s physical activity. For example, you could interpret the data with a sentence like: “Children whose parents had higher education levels exhibited greater time in moderate-to-vigorous physical activity, p = ..., r = ....”
AUTHORS:
Again, the SES in the present study was not one main construct to be studied, but it was a co-variate instead (which usually has impact on the aforementioned relationship, and, for that it was controlled for); the authors kept that clarification throughout the manuscript.
Thus, the authors really would like to clarify that the main purpose of the study was twofold: i) to compare MVPA levels and sedentary time of urban and non-urban children and assess compliance of active children with the PA recommendation; ii) to analyze associations between the risk of overweight and MVPA of children by their degree of urbanization.
In summary, the pragmatism and focus on those aims needed to be kept, especially to emphasize the pertinence of the research question, as well as trying to overcome the gap on this scientific field.
REVIEWER #2:
- Discussion:
Please include numerical data when stating that certain variables are significantly higher or lower.
AUTHORS:
Thank you for the note. The reviewer’s suggestion was considered to improve the manuscript, and some rates of the present study were added; at the same time, we tried to be as balanced as possible to avoid redundancies with tables; hopefully your understanding.
REVIEWER #2:
- Conclusions:
The conclusions are general and could benefit from specific direction. For example: “Interventions to promote physical activity should be tailored to the characteristics of each residential environment, taking into account the limited time spent in moderate-to-vigorous physical activity observed in urban children.” Additionally, a statement about the practical implications of this study should be added. For instance: “The results of this study could guide the development of customized school physical education programs.”
AUTHORS:
Very good point. The manuscript was edited according to the reviewers’ suggestions in its conclusion as practical implications. The authors have added the following content:
“To enhance the practical implications of the present study for child health promotion, particularly in urban settings where children exhibit lower levels of MVPA and higher SB, it is crucial to adopt a multifaceted approach. First, community-based programs should be developed that encourage active play and physical activities tailored to urban environments, such as diverse organized sports programs or after-school programs that promote structured active behavior. Second, schools should integrate more physical education into their curricula and create safe spaces for children to engage in physical activities during recess. Third, public health campaigns must focus on educating families about the importance of reducing screen time and promoting active lifestyles, particularly during weekends when SB is notably higher. Additionally, collaboration with local governments to improve urban infrastructure—such as creating more parks and safe walking paths—can facilitate increased physical activity among children. Finally, ongoing research should be conducted to monitor the effectiveness of these interventions and adapt them based on the unique needs, particularly on those with higher density people.”
The authors also thank the reviewer for his careful reading and suggestions which were considered and indubitably contribute to make the manuscript clearer and in a better standard for the potential reader. Thank you very much.
Round 2
Reviewer 1 Report
Comments and Suggestions for Authors
The similarity rate (31%) is too high; please reduce it.
Comments on the Quality of English Language
I am not qualified to assess the quality of English in this paper.
Author Response
2025 February, 6th
Manuscript ID: healthcare-3407180_R2, entitled " The risk of pediatric overweight and its association with the children’ objectively measured physical activity and sedentary behaviors by area of residence"
The authors thank the reviewers for their thoughtful and construction comments and suggestions. The comments were carefully read, and the revised manuscript was performed accordingly. Responses to specific comments are indicated below in BLUE font.
REVIEWER #1
REVIEWER #1:
The similarity rate (31%) is too high; please reduce it.
AUTHORS:
The authors thank the reviewer, and the authors did their best effort to revise again this manuscript to avoid some overlapping content which clearly comes from the literature produced by our research team (especially at the methods section). Thus, the similarity was clearly reduced according to the reviewer’s suggestion.
In addition, it should be clearly noted that the majority of overlapping content is from our research projects which it is very difficult to be changed, since it is related to the methodological procedures which were similar to other studies of our research team. Of note, they are always cited with the corresponding reference.
Reviewer 2 Report
Comments and Suggestions for Authors
All requirements have been fulfilled. I believe the research paper is now suitable for publication.
Author Response
2025 February, 6th
Manuscript ID: healthcare-3407180_R2, entitled " The risk of pediatric overweight and its association with the children’ objectively measured physical activity and sedentary behaviors by area of residence"
The authors thank the reviewers for their thoughtful and construction comments and suggestions. The comments were carefully read, and the revised manuscript was performed accordingly. Responses to specific comments are indicated below in BLUE font.
REVIEWER #2:
All requirements have been fulfilled.
I believe the research paper is now suitable for publication.
AUTHORS:
The authors really would like to thank the reviewer again for their thoughtful and construction comments and suggestions at the first round of the reviewer process. In fact, all suggestions were considered, and they indubitably contribute to make the manuscript clearer and in a better standard for the potential reader. Thank you very much.